# Two-Stage Recognition Mechanism of the SARS-CoV-2 Receptor-Binding Domain to Angiotensin-Converting Enzyme-2 (ACE2)

**DOI:** 10.3390/ijms25010679

**Published:** 2024-01-04

**Authors:** Iga Biskupek, Artur Gieldon

**Affiliations:** Faculty of Chemistry, University of Gdansk, ul. Wita Stwosza 63, 80-308 Gdansk, Poland; igabiskupek@gmail.com

**Keywords:** SARS-CoV-2, infection mechanism, intermediate state, UNRES, molecular dynamics, molecular modeling

## Abstract

The SARS-CoV-2 virus, commonly known as COVID-19, occurred in 2019. It is a highly contagious illness with effects ranging from mild symptoms to severe illness. It is also one of the best-known pathogens since more than 200,000 scientific papers occurred in the last few years. With the publication of the SARS-CoV-2 (SARS-CoV-2-CTD) spike (S) protein in a complex with human ACE2 (hACE2) (PDB (6LZG)), the molecular analysis of one of the most crucial steps on the infection pathway was possible. The aim of this manuscript is to simulate the most widely spread mutants of SARS-CoV-2, namely Alpha, Beta, Gamma, Delta, Omicron, and the first recognized variant (natural wild type). With the wide search of the hypersurface of the potential energy performed using the UNRES force field, the intermediate state of the ACE2–RBD complex was found. R403, K/N/T417, L455, F486, Y489, F495, Y501, and Y505 played a crucial role in the protein recognition mechanism. The intermediate state cannot be very stable since it will prevent the infection cascade.

## 1. Introduction

The COVID-19 pandemic, caused by the SARS-CoV-2 virus, started in December 2019 [1] and rapidly spread in many Asian countries. After the initial months, the viral dissemination also included Europe and America [2]. In March 2020, COVID-19 was declared as a pandemic by the World Health Organization (WHO). It is one of the most investigated pathogens in human history since (according to Google Scholar) almost 200,000 scientific papers appeared in the last 3 years [3,4]. In comparison, about 81,000 papers about flu were published (mostly in the content of SARS-CoV-2) at the same time. SARS-CoV-2 is a single-stranded positive-sense RNA virus. The central part of the genome is the 5′ terminal, which is rich in open reading frames and encodes proteins essential for virus replication. The 3′ terminal contains the five structural proteins: the spike protein (S), membrane protein (M), nucleocapsid protein (N), envelope protein (E), and haemagglutinin–esterase (HE) protein [5]. The S protein mediates an attachment and fusion between the virus and host cell membrane. The top part of the S protein, called the receptor-binding domain (RBD), links to the human cell via the Angiotensin-Converting Enzyme-2 (ACE2). The S protein in virus-producing cells can be activated by transmembrane serine protease 2 (TMPRSS2), which can cleave the cysteine bridge in receptor-binding domain (RBD), the S1/S2 (SLLR667↓) cleavage site, and additional peripheral K/R residues, and also an unknown site within the S2 domain. This process induces the necessary conformational changes leading to virus–host fusion at the plasma membrane [6]. Numerous theoretical papers on the virus spike receptor and Angiotensin-Converting Enzyme-2 (ACE2) receptor interaction were published since Wang et al. published the crystal structure of the C-terminal domain of the SARS-CoV-2 (SARS-CoV-2-CTD) spike (S) protein in complex with human ACE2 (hACE2) (PDB (6LZG)) [7]. One of the most impressive studies was conducted by Casalino et al. The authors simulated the entire SARS-CoV-2 viral envelope, containing 305 million atoms, using CHARMM36 all-atom additive force fields and NAMD 2.14 with a TIP3P water environment. The simulation was driven by AI-based methods to drive/enhance the conformational sampling of the system. One of the most important pieces of feedback from this paper was the information on the intrinsic flexibility of the virus spike protein that facilitates the interaction with the ACE2 receptors exposed on the host cell [8]. Another paper in which the authors described the motions of the spike receptor was written by Yu et al. The authors used a coarse-grained (CG) model to simulate the SARS-CoV-2 virion. Only the spike protein was able to move, while the rest of the virion proteins were treated as rigid bodies. Principal component analysis was used to identify the spike protein motions, which are crucial in the target recognition process [9]. Similar results were obtained by Ma et al. The authors used the MARTINI model and simulated the complex of the ACE2–RBD (Receptor-Binding Domain) from the spike protein. They pointed to three residues from the RBD, namely F486, Q498, and Y505, which contribute the most to the free energy of binding between the ACE2 and SARS-CoV-2 RBD and play a key role in the initial recognition of the ACE2 by the RBD. The authors concluded that the flexible region of the RBD leads to a higher possibility of binding to ACE2 PD and provides a greater binding affinity [10]. In 2022, Pak et al. performed a coarse-grained molecular dynamics simulation of the SARS-CoV-2 spike receptor. The authors concluded that binding to the ACE2 induces a cyclical tightening and loosening process that progressively weakens S1 interactions and that the process is generally sequential. At first, RBDs conformationally vary between the open and closed state. When they are successfully bound to the ACE2, RBDs are taking the open state [11]. Experimental evidence for sequential steps in the ACE2 binding of the SARS-CoV-2 spike protein was given by Benton et al. two years earlier using cryo-electron microscopy. They pointed out that the geometry of the RBD in a close conformation is incompatible with ACE2. Successive RBD opening and ACE2 binding lead to a fully open and ACE2-bound form [12]. Very similar research was published by Yuan et al. on SARS-CoV and MERS [13]. Research on the balance between the open and closed conformation of the receptor-binding domain was performed by numerous scientific groups and published widely in the literature [14,15,16,17,18,19].

Coronavirus spike protein is a multifunctional molecular machine that mediates coronavirus entry into host cells. In the prefusion conformation, the S1 subunit of the spike protein consists of four domains—the amino-terminal (N-terminal) domain (NTD), the receptor-binding domain (RBD), and two carboxy-terminal (C-terminal) domains (CTD1 and CTD2). The spike protein binds to the target cell via interaction between the ACE2 and RBD, mediating viral attachment to the host [20,21]. This step triggers the next one driven by the S2 subunit–membrane fusion. It is well documented that the spike protein exists in two structurally distinct conformations, prefusion and post-fusion. In the literature, they are called ‘up’ for a receptor-accessible state and ‘down’ for a receptor-inaccessible state. In the post-fusion state, conformational changes lead to S1 subunit disengagement from S2, and likely its dissociation from S2, while S2 undergoes a cascade of refolding events to form a stable and elongated trimer, finally leading to a membrane fusion [20,21,22]. As a consequence, the RBD constantly switches between a standing-up position for receptor binding and a lying-down position. This process is necessary to avoid detection from the target immune system [13,23,24].

Since the RBD is in a constant flux between open and close conformation [25], and it is incompatible with the ACE2 in a close form [12], we can speculate that the SARS-CoV-2 cell recognition mechanism is much more complicated. The final conformation of the RBD–ACE2 complex is well documented in the literature [26,27,28,29,30,31,32]. A very interesting article on the SARS-CoV-2 RBD–ACE2 complex was published by Han et al. [33]. The authors performed a detailed analysis of the most spread SARS-CoV-2 variants (from Alpha to Omicron). They demonstrated that the K417N mutation, responsible for the destruction of the salt bridge with D30_(ACE2)_, is not crucial for the cellular recognition process. For Omicron RBD, the impact of K417N and N501Y substitutions has already been analyzed [32,34]. Surprisingly, Omicron showed no significant change in binding affinity when compared to the other variants and also demonstrated an increased trend in infection efficiency. Han et al. pointed out that the aromatic ring of the Y501 could make new favorable non-bonded interactions with ACE2, such as cation-π interaction with K353. Additionally, the substitution of S477 for N477 on the Omicron RBD confers two new H bonds, with S19 strengthening the ACE2–RBD complex. In other words, some substituted residues decrease the binding affinity between the RBD and ACE2 and others enhance the binding affinity [30]. In most cases, the substitutions and specific deletions were present in the S1/S2 domains and RBD areas of the SARS-CoV-2 genome. As a result, the new variants were identified. Major (non)synonymous mutations affecting the RBD region in novel SARS-CoV-2 include N501Y, E484K, L452R, and K417N/T [35,36]. The SARS-CoV-2 mutation process is driven by the action of natural selection, which is multidimensional. It depends on many factors: suitable steps in the replication process, the response of the immune system, the spread rate, the ability of the host recognition, and many others [37,38]. With the SARS-CoV-2 ACE2 RBD complex analysis, we must be aware that this is only one small piece of the big molecular machinery.

For this paper, we performed a detailed analysis of the ACE2–RBD complex with the hypothesis of the existence of the intermediate state between closed and fully attached RBD and ACE2. With our previous manuscript, we already successfully identified it [32], and using this information, we successfully explained the capability for the infection of the selected species by the SARS-CoV-2 virus. This time, we tried to answer the question of the role of the intermediate state(s) in the RBD–ACE2 recognition process.

## 2. Results

With the information from our previous research [32], a stable intermediate state of RBD–ACE2 complex was identified. At first, all obtained stable intermediate states were compared with each other to verify the relative orientation of the RBD with respect to the ACE2 in all simulated models. The comparison showed that wt, Alpha, and Beta versions of SARS-CoV-2 exhibit very similar conformations while Gamma, Delta, and Omicron were slightly different. This result is consistent with the publication from Mandal et al. [39]. The authors pointed out that stable conformations of different versions of the RBD with ACE2 also differ from each other. Therefore, there is no reason why it should not be true in the case of the intermediate conformation. In most of the performed simulations, we observed the same behavior in the computed system when only the local search was performed. However, a significant percentage of simulations ended differently (see Figure 1 and Table 1). Therefore, we concluded that the UNRES force field was able to identify the second spot on the ACE2 surface putatively responsible for the first step of binding the RBD to the ACE2. It should be noted here that the presented statistic differs from our previous research. This fact is due to the simulation method. In the previous paper, the simulations were performed without any restraints on the protein structure. This caused a more intensive conformational search, and as a result, more simulations ended far away from the starting point. Here, to prevent structural deformation, torsional restraints were added. As a consequence, only a local search was performed in a much larger number of simulations. The other reason for such a variety of conformations is that the difference between calculated binding energies, calculated with an all-atom force field, is within 250 kcal/mol [35].

Statistically, the strongest interaction between the RBD and ACE2 was in the Alpha mutant. In this case, only the local search was performed in about 73% of MD trajectories. On the other side, the weakest interaction between the RBD and ACE2 was observed in the Omicron mutant, in which 60% of trajectories ended near the initial state. If we take a closer look at the article by Han et al. [32], the K_d_ value of the Omicron variant was the largest. The salt bridge between D30_(ACE2)_ and K417_(RBD)_, which was crucial in the wt variant of the virus, can be replaced by the other interactions, albeit not completely.

To explain the observed differences in the locations of the intermediate state, a detailed analysis of the residues located on the protein–protein interface was performed. Eight residues from the Receptor-Binding Domain (RBD), as present in almost every computed system, were identified, namely R403, K/N/T417, L455, F486, Y489, Y495, Y501, and Y505 (see magenta residues in Figure 2). Only two of them are not hydrophobic.

The only possibility for a strong interaction with the residues available in all computed systems is via a salt bridge between R403_(RBD)_ and an acidic residue located on the ACE2 surface. The only system in which such a situation was observed was α-RBD, where R403_(RBD)_ was interacting with E35_(ACE2)_. See the red spot in Figure 2α. The other observed interaction was that of a cation–π type and/or hydrogen bond with tyrosine (observed in wt, α, β, δ, and o) and a hydrogen bond with glutamine residue in δ RBD. In general, R403 was located near hydrophobic residues (see the bright spots in Figure 2: wt, α, β, δ, and o) with the possibility for the creation of a slat bridge, hydrogen bond, and cation–π interaction. The second residue, which can play an important role in the RBD–ACE2 recognition / binding mechanism, is K/N/T417 (see Table 2). In the wt, α, and δ mutants of SARS-CoV-2 RBD in position 417, lysine residue could be found. In the wt mutant, a hydrogen bond with T20_(ACE2)_ and Q24_(ACE2)_ was found, while in the other two variants (α and δ), the salt bridge was present (E23_(ACE2)_ and Q87_(ACE2)_, respectively). The rest of the residues of RBD, found in all of the analyzed systems, namely L455, F486, Y489, Y495, Y501, and Y505, created π–π and hydrophobic interactions with the following residues located on the ACE2 surface: F28, F72, L79, Y83, and L85. In the experimental structure (PDB ID 6LZG), Y495_(RBD)_ together with F497_(RBD)_ was directed towards the protein interior; however, Y495_(RBD)_ was present in all of the computed systems in the intermediate state and F497_(RBD)_ was present in one (see Table 2), being involved in the stabilization of the intermediate state.

With the detailed analysis of the protein–protein interface, we were able to select the RBD fragment (483–493) that was “sticking” to the ACE2 surface and responsible for a good protein–protein adaptation. However, one of the most important residues that did not interact with ACE2 was P491_(RBD)_. It was located at the N-terminal in one of the β-sheets, as visible in the experimental structure (PDB ID: 6LZG), playing a crucial role in the ACE2–RBD interaction. Its function seems to be to adjust the proper conformation by the 478–484 fragment of the RBD with the vicinal protein fragments (especially the residues in positions 484, 486, and 489; see Table 2). Together with the small β-sheet (487–490), those fragments seem to play an important role in the stabilization of the identified intermediate state. Almost all pointed residues are also involved in the interactions identified in the final structure (PDB ID: 6LZG) (see Table 2). The 478–484 fragment of the RBD was sensitive to one general mutation and two additional mutations in the Omicron variant (see Table 2). The most significant mutation was in position 484 (E→A in the Omicron mutant), with the capability of the creation of the salt bridge. In the experimental structure, the acidic and base residue can find a partner for the creation of a salt bridge interaction (K31_(ACE2)_ or E75_(ACE2)_) to reinforce the interaction. In the intermediate state, the role of E/K 484 seems to be not very clear. In the Omicron mutant of RBD, alanine residue is present, and it is directed towards F490_(RBD)_. However, a closer look at the RBD sequence showed that glycine residue (at position 485) in the vicinity was present. As a consequence, we concluded that with the increased conformational flexibility, a better adaptation of the interacting loop was possible. If we take a closer look at Figure 2, the strong interactions like salt bridges and hydrogen bonds are marginal. Hydrophobic interactions, together with π–π, play a major role in the RBD–ACE2 interface. Together with the good conformational adaptation of the RBD, they are key to the molecular recognition of ACE2. With this result, we may speculate that the 474–488 fragment of the RBD plays a crucial in the ACE2 recognition process. Due to its flexibility, a variety of the intermediate conformations were observed (see Figure 1).

To see how the energy profile changes during the simulation and verify the hypothesis of the existence of the second stable minimum on the ACE2 surface, the potential energy plots were drawn.

All of the created energy plots were very similar, with a high jump from the conformation constructed on the PDB: 6LZG template with a local search around the minimum. However, a significant number of conformations (see Table 1) ended in a specific and stable intermediate state with a low energy decrease in some parts of the MD trajectory. This is not very clearly visible on the potential energy profile (see Figure 3), and therefore, we decided to calculate an interaction energy. It should be noted here that due to the parameterization procedure applied in the UNRES force field, the free energy term is also included in the potential energy function. In the local search trajectories, fluctuations of around 5 to 7 Å from the starting structure were observed. In the MD in which a much wider search was performed, the situation was going through some unspecified number of intermediate states to stay for some time on the top of the ACE2. In the case of the wt system, it was around 11 to 13 Å.

The lowest interaction energy, calculated for the initial structure, was observed in the case of the Alpha mutant (see Figure 4a). If we compare this with the work of Han et al. [33], the values fit well since this mutant showed the lowest dissociation constant. On the other side, there is an Omicron mutant with the highest dissociation constant and the highest observed interaction energy. The rest of the predictions also hold well; however, the interaction energy calculated for Beta and Gamma mutants seems to be a little overestimated. If we compare the difference between the interaction energy calculated for the initial structure and calculated in the intermediate state, the largest values were in the cases of the Alpha and Delta mutants (80–100 kcal/mol) and the smallest in the case of Gamma and Omicron (about 40 kcal/mol). Beta and Delta preserved a similar difference (about 70 kcal/mol). The most surprising result was in the cases of Delta and Omicron variants, with the lowest difference between two identified conformations. This suggests that the newly found intermediate state has almost the same interaction energy and is highly probable.

## 3. Discussion

The results presented here allowed us to speculate on the putative Receptor-Binding Domain (RBD)–Angiotensin-Converting Enzyme-2 (ACE2) binding/recognition mechanism. The pointed proteins are needed only in one of the first steps of the virus cell recognition process, and both are finally detached. Therefore, the most important process is to fit one protein to another. The probability of hitting and sticking to the optimal conformation directly is rather small; therefore, the existence of the intermediate state(s) is highly probable. This hypothesis is confirmed by the experimental data since the closed form of the RBD did not fit to the ACE2. The location of the newly found spot on the ACE2 surface is well accessible for attack. The energy minimum in the intermediate state cannot be very deep since the infection process will stop in the wrong place (see our previous work [32]). As a consequence, mostly, the hydrophobic contacts (R403, K/N/T417, L455, F486, Y489, Y495, Y501, and Y505) in the intermediate state were identified. Those contacts seem to be key to the protein–protein recognition process. A variety of the conformations observed in the intermediate state can be explained by the flexibility of the 478–484 RBD fragment and additionally by the presence of G485 and A484 in the Omicron variant. A rather flat hypersurface of the potential energy and well-accessible local minimum can explain why SARS-CoV-2 was so successful in the cell recognition process.

## 4. Materials and Methods

The coronavirus spike receptor-binding domain, complexed with its receptor ACE2 (PDB code 6LZG) [7], was used as a template for modeling. Alpha (B.1.1.7), Beta (B.1.351), Gamma (P.1), Delta (B.1.617.2), and Omicron (B.1.1.529) variants [40] were modeled (see Table 1). To obtain the models of the examined RBD–ACE2 complexes, the point mutations in the receptor-binding domain were made using the UCSF Chimera 1.17.1 software [41] (see Table 1). The most probable or the most fitting rotamer was applied during the computer mutation with the rotamer library, as implemented in the UCSF Chimera 1.17.1 software [42].

The newly built variants (α, β, γ, δ, and o) of the RBD of the SARS-CoV-2 spike receptor, complexed with ACE2 models, were simulated in the UNRES forcefield. UNRES is a united-residue forcefield designed to simulate peptides and proteins. The protein chain is represented by a sequence of Cα atoms, connected via virtual bonds and the united peptide groups located in the middle, with attached united sidechains. UNRES force field has proven its capability for searching the conformational space of the proteins even without known experimental data [43]; therefore, it seems to be a much more useful tool for the searching of the intermediate states than all-atom force fields since UNRES is capable of the prediction of the protein folding pathway. Forty independent molecular dynamics runs were carried out for five constructed complexes. To prevent the protein from unfolding, virtual angle restraints were added. Ten million MD steps of 1 MTU (molecular time unit) were computed. Snapshots were taken every 10,000 steps. The MTU used in UNRES MD amounts to 48.9 fs, which leads to 48.9 ns of the simulation time of each trajectory. However, it should be noted that because of averaging over the secondary degrees of freedom, the time scale of UNRES MD is extended by 1000–10,000 times compared to the all-atom time scale. The discrepancy between the simulation and the biological time scale restricts the applicability of this technique for solving concrete biological problems [44,45,46]. The final models were converted into all-atom representations using the PULCHRA [47] method and subjected to short energy minimization with the AMBER force field [48]. It should be noted that one of the disadvantages of the coarse-grain force field is that after the simulation, the reconstruction process of the all-atom model is required. In this way, some information on the sidechain conformation is lost. Since the interaction energy calculation is not implemented in UNRES, it was performed manually. Three of the lowest potential energy structures from separate MD runs were selected and single-point energy was calculated. In the next step, RBD and ACE2 were separated, and again, single-point energy was calculated. The difference is shown in Figure 4.

The electrostatic surface was calculated using APBS, the adaptive Poisson–Boltzmann solver [49], as implemented in the PyMol (Delano Scientific, San Carlos, CA, USA) plugin. The continuum electrostatics calculations used for the surface as drawn were calculated using the PDB2PQR 3.6.1 software [50].

## 5. Conclusions

One of the possible ways to catch protein intermediate states is via computer simulation [51]. The protein–protein association problem seems to be even more complicated since, in the literature, some arguments for a rapid association and association through some number of intermediate states can be found [52]. Here, we demonstrated the existence of the intermediate state(s) of the angiotensin-converting enzyme-2 and SARS-CoV-2 receptor-binding domain (RBD–ACE2) complex. Since the SARS-CoV-2 threat is still very high (according to the WHO dashboard on 25 October 2023, there have been 771,549,718 confirmed cases of COVID-19, including 6,974,473 deaths), any additional information on the infection mechanism can be crucial for the full understanding of the infection process. Here, we proposed the existence of the intermediate state on the receptor-binding domain (RBD) of SARS-CoV-2 and the human Angiotensin-Converting Enzyme 2 (ACE2) recognition pathway. Mostly, the hydrophobic residues, namely L455, F486, Y489, Y495, Y501, and Y505, together with good conformational adaptation, play a key role in the first step of the RBD–ACE2 contact initiation. The good conformational adaptation is caused by the presence of P491(RBD) and G485(RBD), additionally reinforced in the Omicron variant by R484→A (RBD) mutation. However, in our study, together with the K417→N mutation in the Omicron RBD, the interaction in the final RBD–ACE2 conformation was weakening.

## Figures and Tables

**Figure 1 ijms-25-00679-f001:**
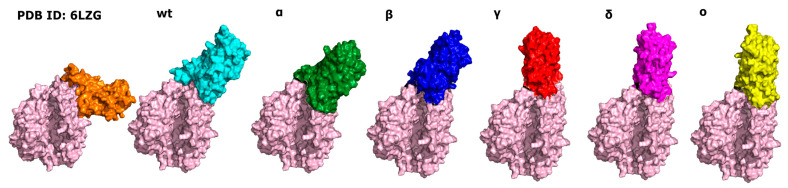
The comparison of the starting point (orange) with the identified intermediate state in all computed systems. The shape of the ACE2 protein was marked in light pink. Wild-type RBD taken from PDB (ID: 6LZG) was marked orange and cyan in the stable and intermediate conformations, respectively. The intermediate states of the computed versions of SARS-CoV-2 RBD, Alpha, Beta, Gamma, Delta, and Omicron, were marked as follows: green, blue, red, violet, and yellow.

**Figure 2 ijms-25-00679-f002:**
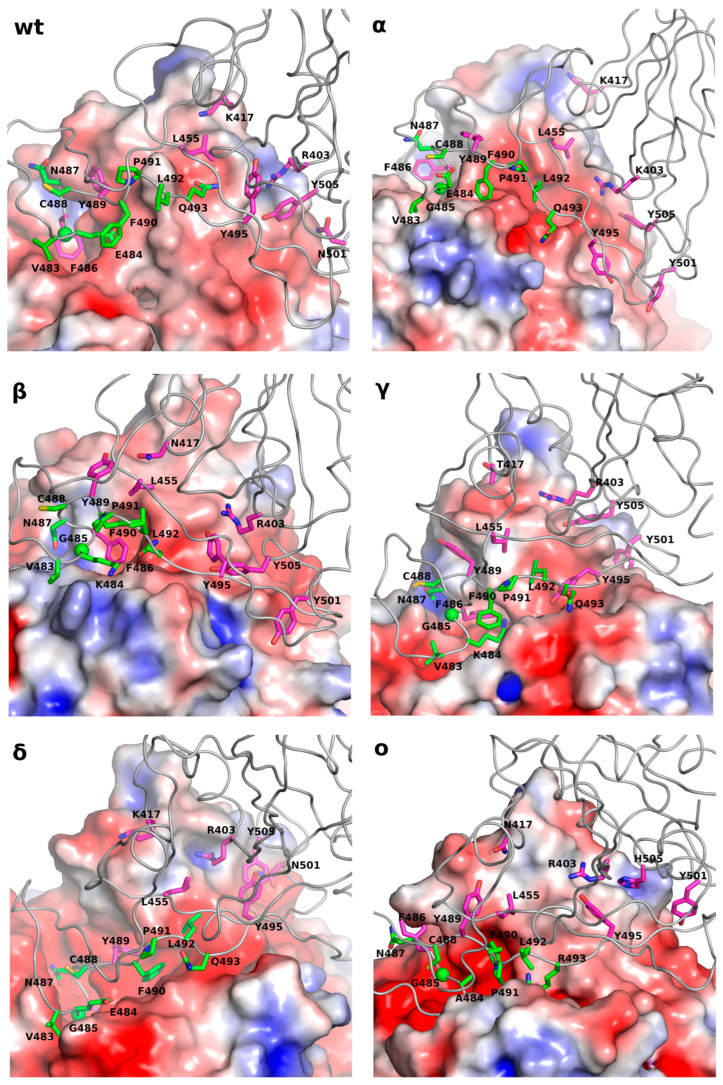
The intermediate states, identified in all simulated variants of the Receptor-Binding Domain together with Angiotensin-Converting Enzyme-2, were labeled with Greek letters. The most important residues identified as playing a crucial role in the first step of the RBD-ACE2 recognition were labeled in magenta. The residues located in the loop that “sticks” to ACE2 were labeled in green. The Poisson–Boltzmann Surface of the ACE2 protein was generated by using ABPS as implemented in Pymol 2.5.5 software (see methods). The blue region shows the location of positive electrostatic potential, while the red region is the location of negative electrostatic potential. The source of the red spots located in the binding grove comprised the oxygen atoms from the peptide bonds.

**Figure 3 ijms-25-00679-f003:**
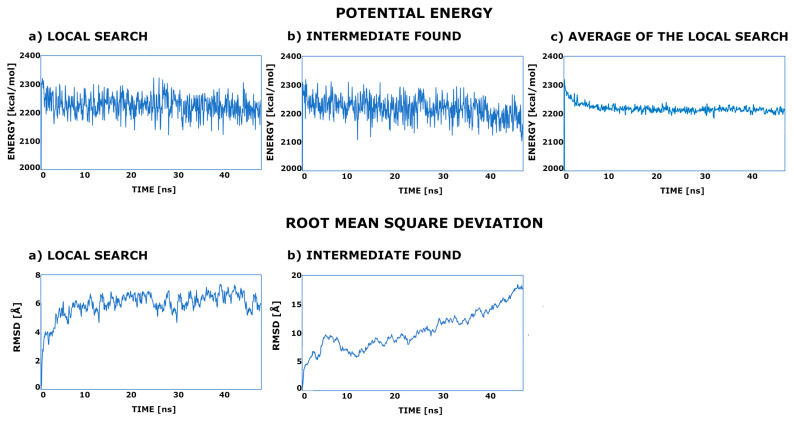
The potential energy and root mean square deviation plots of the selected MD trajectories of the wt variant: (**a**) the MD trajectory in which only the local search was performed; (**b**) the MD trajectory after having led to the intermediate state; (**c**) the average energy calculated for all simulations in which only a local search was performed.

**Figure 4 ijms-25-00679-f004:**
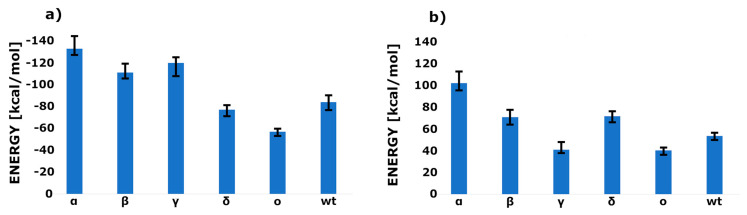
The interaction energy plots for all computed RBD–ACE2 systems. (**a**) The interaction energy of the initial structures, constructed using PDB ID: 6LZG experimental structure as a template. (**b**) The energy difference between the identified intermediate state and the initial structure.

**Table 1 ijms-25-00679-t001:** The statistical analysis of the performed simulations. The numbers show the number of simulations that ended in a selected state. ‘Initial’ means that during the simulation, the computed system remained unchanged and only a local search was performed. ‘Intermediate’ means that the simulation ended in the intermediate state. The sum is not equal to 100% since some of the simulations led to a different state.

	wt	α	β	γ	δ	o
initial	70%	73%	68%	70%	63%	60%
intermediate	20%	25%	30%	28%	30%	25%

**Table 2 ijms-25-00679-t002:** Summary of mutations in individual amino acid residues of the receptor-binding domain in selected variants of the SARS-CoV-2 virus [40], together with the residues identified in the intermediate state of RBD–ACE2 complex. The first identified version of SARS-CoV-2 was called wt (wild type). The column labeled wt-6LZG contains residues located on the RBD–ACE2 interface taken from the experimental structure (PDB ID: 6LZG).

Mutations Identified in the RBD					Residues Located on the RBD–ACE2 Interface in the Intermediate State						
α	β	γ	δ	o	wt-6LZG	wt	α	β	γ	δ	o
				339, G→D							
				371, S→L							
				373, S→P							
				375, S→F							
						R403	R403	R403	R403	R403	R403
								D405	D405	D405	
											E406
									R408		
								Q409			
								Q414			
								T415			
	417, K→N	417, K→T		417, K→N	K417	K417	K417	N417	T417	K417	N417
							I418	I418			
							Y421				
				440, N→K							
				446, G→S							
					N478			N448	N448	N448	N448
			452, L→R								
					Y453		Y453				Y453
					L455	L455	L455	L455	L455	L455	L455
					F456	F456	F456				
								R457	R457	R457	R457
							T470			T470	
					Y473			Y473			
				477, S→N				S477			
				478, T→K					T478		
									N481	N481	N481
						V483	V483			V483	V483
	484, E→K	484, E→K		484, E→A	E484	E484	E484	K484	K484	E484	A484
					F486	F486	F486	F486	F486	F486	F486
			487, T→K		N487	N487		N487			N487
					Y489	Y489	Y489	Y489	Y489	Y489	Y489
							F490		F490	F490	F490
							P491				
						L492	L492			L492	L492
				493, Q→R	Q493	Q493					R493
						S494		S494		S494	
						Y495	Y495	Y495	Y495	Y495	Y495
				496, G→S							
							F497				
				498, Q→R	Q498	Q498			Q498	Q498	R498
					T500					T500	
501, N→Y	501, N→Y	501, N→Y		501, N→Y	N501		Y501	Y501	Y501	Y501	Y501
								V503			
				505, Y→H	Y505	Y505	Y505	Y505	Y505	Y505	H505

## Data Availability

Data are contained within the article.

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
