# Peer review of "Two-Stage Recognition Mechanism of the SARS-CoV-2 Receptor-Binding Domain to Angiotensin-Converting Enzyme-2 (ACE2)"

_ijms, 2024, doi:10.3390/ijms25010679_

Round 1

Reviewer 1 Report

Comments and Suggestions for Authors

The aim of this manuscript is to perform a detailed analysis of the ACE2-RBD complex with the hypothesis of the existence of the intermediate state between closed RBD to ACE2.

This manuscript shows rich content, providing a deep insight for some works: the study is within the journal’s scope, and I found it to be well-written, providing sufficient information. Even if the manuscript provides an organic overview, with a densely organized structure and based on well-synthetized evidence, there are some suggestions necessary to make the article complete and fully readable. For these reasons, the manuscript requires major changes.

Please find below an enumerated list of comments on my review of the manuscript:

MINOR POINTS:

The authors should provide a list of the abbreviations, mentioned in this manuscript.

ABSTRACT:

LINE 12-13: Please, remove the space, between these sentences.

LINE 14: The authors should rewrite this sentence in a more fluent way as following: “The aim of this manuscript is to simulate the most widely spread mutants of COVID-19, namely: alpha, beta, gamma, delta, omicron, and the first recognized variant (natural-wild type)”.

MAJOR POINTS:

INTRODUCTION:

LINE 23: The COVID-19 pandemic started in the seafood market of Wuhan, China, in early December 2019 and then rapidly spread to Thailand, Japan, South Korea, Singapore, and Iran. After these initial months, the viral dissemination included Italy, Spain, the USA, the UAE, and the U.K (see, for reference: Umakanthan, S.; Sahu, P.; Ranade, A.V.; Bukelo, M.M.; Rao, J.S.; Abrahao-Machado, L.F.; Dahal, S.; Kumar, H.; Kv, D. Origin, transmission, diagnosis and management of coronavirus disease 2019 (COVID-19). Postgrad. Med. J. 202096, 753–758). In this introductive section, the authors should describe the diffusion of COVID-19 pandemic, as a global health threat, as declared by the World Health Organization (WHO) on 11 March 2020.

LINE 26: The causative agent for COVID-19, severe acute respiratory syndrome coronavirus-2 (SARS-CoV-2), is an enveloped positive single-stranded RNA virus, whose viral genome is characterized by a 5’ terminal, rich in open reading frames, which encodes proteins fundamental for virus replication. Furthermore, the 3’ terminal includes five structural proteins, Spike protein (S), membrane protein (M), nucleocapsid protein (N), envelope protein (E), and hemagglutinin-esterase protein (HE). The Spike protein is responsible for the pathogenesis in the human species since its receptor-binding domain (RBD) links to human cell surface receptor protein Angiotensin-Converting Enzyme-2 (ACE-2), encoded by the ACE2 gene. Hence, the virus, through the transmembrane protease serine 2 (TMPRSS2), a cell–surface protein expressed by epithelial cells of specific tissues, is uploaded to the tissues (see, for reference: Torge, D.; Bernardi, S.; Arcangeli, M.; Bianchi, S. Histopathological Features of SARS-CoV-2 in Extrapulmonary Organ Infection: A Systematic Review of Literature. Pathogens 202211, 867. https://doi.org/10.3390/pathogens11080867). This is the major concern of this manuscript, which may benefit from providing recent evidence on the molecular features of SARS-CoV-2 infection.

The main topic is interesting, and certainly of great clinical impact. As regards the originality and strengths of this manuscript, this is a significant contribute to the ongoing research on this topic, as it extends the research field on the analysis of the ACE2-RBD complex with the hypothesis of the existence of the intermediate state between closed RBD to ACE2. Overall, the contents are rich, and the authors also give their deep insight for some works.

As regards the section of methods, there is a specific and detailed explanation for the methods used in this study: this is particularly significant, since the manuscript relies on a multitude of methodological and statistical analysis, to derive its conclusions. The methodology applied is overall correct, the results are reliable and adequately discussed.

The conclusion of this manuscript is perfectly in line with the main purpose of the paper: the authors have designed and conducted the study properly. As regards the conclusions, they are well written and present an adequate balance between the description of previous findings and the results presented by the authors.

Finally, this manuscript also shows a basic structure, properly divided and looks like very informative on this topic. Furthermore, figures and tables are complete, organized in an organic manner and easy to read.

In conclusion, this manuscript is densely presented and well organized, based on well-synthetized evidence. The authors were lucid in their style of writing, making it easy to read and understand the message, portrayed in the manuscript. Besides, the methodology design was appropriately implemented within the study. However, many of the topics are very concisely covered. This manuscript provided a comprehensive analysis of current knowledge in this field. Moreover, this research has futuristic importance and could be potential for future research. However, major concerns of this manuscript are with the introductive section: for these reasons, I have major comments for this section, for improvement before acceptance for publication. The article is accurate and provides relevant information on the topic and I have some major points to make, that may help to improve the quality of the current manuscript and maximize its scientific impact. I would accept this manuscript if the comments are addressed properly.

Comments on the Quality of English Language

LINE 12-13: Please, remove the space, between these sentences.

LINE 14: The authors should rewrite this sentence in a more fluent way as following: “The aim of this manuscript is to simulate the most widely spread mutants of COVID-19, namely: alpha, beta, gamma, delta, omicron, and the first recognized variant (natural-wild type)”.

Author Response

Dear Reviewer,
I'm attaching the doc file with all the answers. I hope You will be satisfied.

Sincerely,
Artur Gieldon

Reviewer 2 Report

Comments and Suggestions for Authors

The manuscript by Biskupek and Gieldon describes a coarse-grained molecular dynamics of the binding between RBD and ACE2. Based on the simulation results, the authors proposed that there are intermediate states in the recognition mechanism.

While the conclusion is interesting, it is not strongly supported by the results due to flaws in the research design. Therefore, this work in its current form is not recommended for publication in IJMS.

(1) The biggest issue is that it's not clear whether the observed intermediate states are artifacts of the force field. In all of the simulations, the initial states account for over 60%.  When evaluating a force field for tasks such as the prediction of protein-protein binding, it is generally considered an impressive performance if the force field can predict the correct structure to be most stable for all of the systems, and other less stable structures would be assumed to be caused by inaccuracies of the force field. Therefore, the simulation results presented in this manuscript alone cannot serve as strong evidence for the existence of intermediate states.

(2) The simulations do not provide a timescale for the conversion between the initial and the alternative structures, and it does not prove whether the binding has to go through the intermediate structure.

(3) Justification of the simulation length should be justified. Based on Figure 3b, there seems to be a drift in the energy throughout the simulations. Also, I would recommend a convergence analysis based on the structures, in addition to those based on energy.

(4) Could the authors comment on how the open and closed states of RBD are sampled during the simulations?

(5) It would be helpful to provide error bars in Figure 4. It is not conventional to have a decreasing y-axis in Figure 4a. An alternative may be plotting the initial and final energy side-by-side, rather than initial energy and energy difference.

(6) Please provide more details on how interaction energies are calculated (by using coarse-grained or all-atom force field), and how the intermediate states are identified and how the percentage in Table 1 are calculated. 

Comments on the Quality of English Language

Some of the language needs to be polished. For example, the words "occur" and "appear" were used many times in Abstract and Introduction; Line 62 "We know that ..." seems a little informal. 

Author Response

(The authors gave the same response as above.)

Reviewer 3 Report

Comments and Suggestions for Authors

This is an interesting and a reasonably good research paper. There is still room for improvement.

1) The first 3 sentences of the abstract is irrelevant as everybody knows about SARS-CoV-2 and COVID-19. They should be deleted, and the authors should instead use the extra space to describe previous research and what does this paper offer than is not already done.

2) SARS-CoV-2 and COVID-19 are not synonymous as described on line 23. The former refers to the virus. The other refers to the disease caused by the virus.

3) Line 23 is wrong. COVID-19 did NOT occur in 2019. It was first discovered or noticed in 2019.

4) There is inconsistent usage of "SARS-CoV-2". Sometimes "SARS-CoV-2". Sometimes "SARS-CoV2". Correct and consistent usage is important.

5) The authors like to use the words "and coworkers". A more correct and usual way to denote this is to use the abbreviation "et. al". For example, they can write "Yu et. al" on line 38.

6) There should be more discussion and summary of the strengths and weaknesses of their research as compared to past research.

Comments on the Quality of English Language

The English is fine with the exception of line 23. The authors may not have understood the meaning of "occur" or they could have meant "initially occurred". Even if they had used "initially occurred", it would have been scientifically wrong because there was cryptic transmission of the virus long before 2019. Also, its syntax is not correct. Viruses don't "occur"

Author Response

(The authors gave the same response as above.)

Round 2

Reviewer 1 Report

Comments and Suggestions for Authors

The authors have significantly improved the manuscript.

Author Response

Thank, you.

Reviewer 2 Report

Comments and Suggestions for Authors

The authors addressed some of my concerns in the revision and response letter, while other issues have not been answered.

(1) this work only shows that there are alternative conformations to the initial structure, and it does not provide evidence that the binding event has to go through the intermediate state. To support this, there should be simulations that start from the unbound state, then convert to the intermediate state, and finally to the bound state. Alternatively, assuming the binding is reversible, it could be observation of the unbinding event that pass through the intermediate state.

(2) The simulation length seems insufficient for the estimation of interaction strength (as shown in Table 1). The author mentioned that there are many "boring" simulations that stayed at the initial conformation, which is also suggested by the new plot Fig. 3c. This means that the simulation timescale is not long enough to observe the transition between the two conformations. So the results in Table 1 are likely kinetically controlled. So one can only draw conclusion about kinetics and not thermodynamics (e.g. interaction strength and comparison with kD).

(3) As the author mentioned, there are large fluctuation of the energy, so Fig 3a/b are not easy to interpret. The authors added the average energy of local-search simulations. However I think convergence analysis based on structures, e.g. a plot of RMSDs from the initial structure and from the intermediate structure over time, is more straight forward.

(4) One of my original comments was whether the open and closes states of RBD are observed during the simulation, and how does it relate to the initial and intermediate conformations.

(5) I still insist that it is important to estimate the errors for Figure 4 before interpreting the data, especially since the interaction energy might be noisy. The authors mentioned that there is no implementation of the interaction energy calculation so it was done by hand, but it should be possible to do by using some simple scripts.

(6) Please provide details on how the intermediate conformations are defined: what is the RMSD cutoff or the parameters for clustering analysis if applicable.

(7) The authors suggest in the response letter that the interaction was calculated by UNRES force field. Please include it in the Methods section. 

Author Response

Dear Reviewer,

Please find the replies in the file attached.

Sincerely
Artur Gieldon

Reviewer 3 Report

Comments and Suggestions for Authors

Improvements seen

1) Line 292 "SARS-CoV2" needs to be changed to "SARS-CoV-2"

2) Inconsistent use of the words "Omicron", "Alpha"..etc.. Sometimes "omicron" "delta'..etc.. Sometimes "Omicron", "Delta"..etc... Should be "Omicron" "Delta"..etc

Author Response

Corrected.

Round 3

Reviewer 2 Report

Comments and Suggestions for Authors

First, I'd like to make it clear to the authors that I'm not requesting additional simulations, but the main conclusions need to be toned down since they are not strongly supported by the current results, and limitations of the the methodology should be acknowledged.

(1) I'm aware of the experimental and computational evidence of intermediate state mentioned in the Introduction. However, most of the studies are related to the open and closed conformation of RBD, which is not discussed in the main results of the paper. So the conformations found in this study are not necessarily the same as the intermediate states in previous studies that play a role in binding.

Therefore the evidence provided in this study is indirect, which should be mentioned. Also I think it is helpful to mention that RBD open and closed conformation are observed in trimer, while the simulations are for monomer, and any reason behind this.

(2) If the goal of the simulations is just conformational search, the author should mention that there are more efficient methods available for UNRES force field, such as conformational space annealing and MREMD.

If the goal is mechanistic understanding of the two-stage recognition, then the simulations are clearly not converged, and it should be acknowledged.

(3) The data in Table 1 does not have enough statistical power to decide which is the most stable mutant. For example, the Omicron has a 60% ratio of simulations that stay in the initial conformation, but the 95% CI is roughly 48-73%. So there is no statistically significant difference between Omicron and Sigma, and between wt, alpha, beta and gamma. So it is better not to over-interpret the data.

(4) Line 306: the author mentioned "a high jump from the conformation constructed on the PDB". Does this mean that the energy increases immediately at the start of the simulation? I noticed that in Figure 3c, the energy seems to go from 2000 to 2300, and then gradually decreases to about ~2200. Is this correct?

Line 360: "the difference between the interaction energy calculated for the initial structure and calculated in the intermediate state". Does this mean the interaction energy in the intermediate state is more positive? Please define this more clearly.

If the intermediate state is hundreds of kca/mol more positive than the initial conformation, then please discuss whether it affects the conclusion.

(5) Line 408-410: "Ten million MD steps of 1 MTU (molecular time unit) were computed. The snapshots were taken every 10.000 steps. The MTU used in UNRES MD amounts to 48.9 fs, which leads to 489 ns of the simulation time"

I think the default time step in UNRES is 0.1 MTU = 4.89 fs. Please clarify in the main text whether its 100 million steps of 0.1 mtu, or 10 million steps of 1 mtu, or 10 million steps of 0.1 mtu.

Author Response

Dear Editor / Dear Reviewer,

I’ve prepared a summarized correspondence with the Reviewer. I’m aware that UNRES is still in development, however, it proved the capability of the prediction of the protein structure, which was published in many papers. The most relevant are from the CASP experiment. Since the force field is still in development, some calculations have to be done by hand.

This is a theoretical paper, based on experimental facts. The reviewer pointed, out that we found only another binding mode. Yes, but with all pointed experimental evidence, the existence of such a point is probable.

Since it is a theoretical paper with indirect evidence, in the conclusion we pointed, that, till now, our results are pure speculation, based only on in silico experiments. ”The results presented here allowed us to speculate on the putative Receptor Binding Domain (RBD) – Angiotensin Converting Enzyme-2 (ACE2) binding/recognition mechanism.”

Maybe someone someday will see it in the experiment – I hope.

Sincerely

Artur Gieldon

Round 4

Reviewer 2 Report

Comments and Suggestions for Authors

I could not find the summarized correspondence as mentioned in the cover letter, and I don't see revisions related to some of my previous comments.

I don't know whether it's an error in the submission system or an oversight. I would appreciate it if the authors could resubmit this version and double check the attachment.